# Exogenous NO Therapy for the Treatment and Prevention of Atherosclerosis

**DOI:** 10.3390/ijms21082703

**Published:** 2020-04-13

**Authors:** Tommaso Gori

**Affiliations:** Department of Cardiology, University Medical Center Mainz and Deutsches Zentrum für Herz und Kreislauf Forschung, 55131 Standort Rhein-Main Mainz, Germany; tommaso.gori@unimedizin-mainz.de; Tel.: +496131-172829; Fax: +49-6131-176428

**Keywords:** organic nitrates, nitric oxide, nitric oxide donors

## Abstract

Amyl nitrite was introduced in 1867 as the first molecule of a new class of agents for the treatment of angina pectoris. In the following 150 years, the nitric oxide pathway has been the subject of a number of pharmacological approaches, particularly since when this elusive mediator was identified as one of the most important modulators of vascular homeostasis beyond vasomotion, including platelet function, inflammation, and atherogenesis. While having potent antianginal and antiischemic properties, however, nitric oxide donors are also not devoid of side effects, including the induction of tolerance, and, as shown in the last decade, of oxidative stress and endothelial dysfunction. In turn, endothelial dysfunction is itself felt to be involved in all stages of atherogenesis, from the development of fatty streaks to plaque rupture and thrombosis. In the present review, we summarize the agents that act on the nitric oxide pathway, with a particular focus on their potentially beneficial antiatherosclerotic and unwanted pro-atherosclerotic effects.

## 1. Introduction

Atherosclerosis and its systemic correlates in the coronary, cerebral, and peripheral circulations represents the leading cause of death and disability worldwide [1]. The mechanisms underlying the pathophysiology of atherosclerotic plaques are multifaceted and summarize complex interactions among plasma lipid homeostasis, injury reactions, immunity/inflammation, and multiple types of cells in the vascular wall [2]. The hypothesis that free radicals might have a role in this complex framework was raised as early as 1950 when oxidatively modified lipids and proteins were observed in vascular lesions. This evidence led to the hypothesis that oxidative stress might be involved in the processes leading to atherogenesis and plaque instabilization (reviewed in [3]). Only later, at the end of the XXth century, studies demonstrated that another free radical, nitric oxide (NO), is among the most important mediators of vascular homeostasis, and that this elusive molecule with a very short half-life controls numerous vascular functions, including inhibition of smooth muscle proliferation, vascular inflammation, platelet activation, and vascular tone. Based on this evidence, some authors brought forward the idea that exogenous sources of NO might recapitulate the antiatherosclerotic effects of endogenous NO, a hypothesis that has been supported in some, but not all the literature. In contrast, other findings emphasize the potentially deleterious effects of this “exogenous NO replacement therapy”. The present review focuses on the effects of exogenous NO on atherogenesis, and the potential and limitations of NO-donor supplementation in the prevention of plaque growth and instabilization.

## 2. Vascular Free Radicals and Nitric Oxide

A number of molecular sources of free radicals are present in the vascular wall, and free radicals are important mediators of both vascular homeostasis and vascular pathophysiology [4]. Of these radicals, the ones whose pathophysiological role is best understood include NO and reactive oxygen species (ROS). Under physiological conditions, the NO synthase (NOS) produces NO, with a number of vasoprotective effects for the endothelium and vascular homeostasis in general.

While the major source of NO is the four isoforms of NO synthase (NOS, the “classical” endothelial, neuronal, and inducible nitric oxide synthase and the more recently studied red blood cell NOS, with different tissue specificities and regulation mechanisms), ROS are produced by a number of systems, including NADPH (nicotinamide adenine dinucleotide phosphate) oxidase, xanthine oxidase, enzymes of the mitochondrial respiratory chain, and a dysfunctional, uncoupled NO synthase (NOS) [3,5]. Importantly, the function of these enzymatic systems is itself redox-dependent, and a cross-talk between these mechanisms modulates, along with others, their function. For instance, endothelial NOS may produce superoxide anions in settings of reduced substrate (*L*-arginine) availability but also in case of oxidation of the NOS cofactor tetrahydrobiopterin (BH_4_). In this case, the NOS flavoprotein transfers the electron from NADPH to the heme center of the NOS where a ferric heme-superoxy moiety is formed. In the presence of the reduced cofactor BH_4_, this unstable intermediate donates the second electron to ferric heme-superoxy species, which then reacts with L-arginine or *N*-hydroxyl-*L*-arginine to form NO and *L*-citrulline. In the absence of BH_4_, this species is unstable and decays to release superoxide anions. NOS uncoupling is a major mechanism of endothelial/vascular dysfunction in hypertension, hypercholesterolemia, heart failure, reperfusion injury, diabetes, noise-induced stress, and during nitrate tolerance [5,6,7]. These processes are depicted in Figure 1.

Among the numerous sources of ROS, mitochondrial oxidative phosphorylation is probably the less well understood, but quantitatively not the least relevant. During respiration, mitochondria generate physiological levels of superoxide, normally scavenged to water by the manganese-dependent superoxide dismutase and the glutathione peroxidase 1. An inappropriately high production of ROS from mitochondria can occur under pathological conditions, a phenomenon during which electrons are deviated from the respiratory chain to molecular oxygen, triggering cross-activation of reactive oxygen species production from other cellular sources [5]. 

## 3. The Dual Role of Nitric Oxide Donors

A defect in NO production, or an increased scavenging by ROS, is, therefore, one of the prevalent mechanisms of atherosclerosis. This reduced NO bioavailability leads to vasoconstriction, platelet aggregation, smooth muscle cell proliferation and migration, leukocyte recruitment and adhesion, and lipid deposits in the vascular media in the macro- and microcirculation [8]. 

Because of these protective effects of NO and because of its importance in vascular homeostasis, it might be hypothesized that the exogenous administration of this mediator could be effective in the prevention and/or treatment of atherosclerotic cardiovascular disease. A number of NO donors, direct and indirect, are available. While some among these molecules are spontaneous NO donors, other compounds require enzymatic catalysis or release NO after reacting with reduced thiols. The rapid reaction of NO (whether endogenous or exogenous) with ROS to produce reactive nitrogen species (RNS), however, results in a highly reactive product that, such as ROS, is involved in the inactivation of several enzymatic systems, including those of the NO pathway (Figure 1). Thus, tissue levels of both mediators need to be modulated to preserve NO´s therapeutic benefits while at the same time avoiding its negative effects. This equilibrium, particularly in subcellular microdomains in which ROS are produced, such as mitochondria, is particularly complex to achieve. In sum, NO supplementation might have both positive and negative effects. The impact of different pharmacological approaches targeted at increasing the bioavailability of NO on atherogenesis will be discussed in the following paragraphs. These pathways are presented in the central illustration. 

## 4. Indirect NO Donors

Organic nitrates, including nitroglycerin (GTN), isosorbide mono- (ISMN) and dinitrate (ISDN)] (Figure 2), belong to the cardiovascular pharmacopoeia for 150 years. Principally, these drugs are used for their vasodilator properties in the treatment of symptomatic coronary artery disease and heart failure. In contrast to these standard hemodynamic effects, other non-hemodynamic effects of nitrates have been much less well studied [9,10]. In a series of studies testing the effect of chronic administration of organic nitrates on the progression of atherosclerotic plaques in rodent models, long-term treatment with a low dose of pentaerythritol tetranitrate or isosorbide has been shown to reduce the progression of lesion formation, endothelial dysfunction, and LDL-oxidation [11,12]. Conflicting with these results, however, continuous treatment with nitroglycerin or isosorbide mononitrate caused endothelial dysfunction, increased ROS formation via increased expression and activity of NADPH oxidases, uncoupling of the endothelial NOS due to decreased tetrahydrobiopterin regeneration, and *S*-glutathionylation, all effects that appeared to be at least in part mediated by activation of Angiotensin-1 receptors [13,14]. Further, in vitro, nitroglycerin has been shown to induce uncoupling of the mitochondrial respiratory chain, resulting in increased ROS production [14], triggering a cross-talk with other cellular ROS sources. These adverse effects were less evident in gp91phox knockout mice and were normalized by bosentan, demonstrating a role for increased endothelin-1 production [15]. Finally, in contrast to ISMN, the organic nitrate PETN was associated with an improvement in endothelial responses in a murine model of diabetes, suggesting that this molecule might actually have vasoprotective properties mediated by the reversal of eNOS uncoupling and NADPH oxidase activation [16].

In humans, continuous administration of ISMN and GTN, but not of PETN, has been associated with endothelial dysfunction in both conductance and resistance arteries and with increased production of isoprostanes as evidence of increased oxidative stress [17]. Confirming the oxidative pathophysiology behind these abnormalities, the administration of folic acid, an intervention aimed at restoring the bioavailability of BH_4_, reversed endothelial dysfunction and, additionally, it prevented the development of nitrate tolerance by restoring the oxidative balance of endothelial cells [18]. In contrast, a recent small case-control study from Wei et al. investigated the effect of nitrate esters (taken as a class) on different parameters connected with atherogenesis and inflammation, including infiltration of M1/M2 macrophages, the expressions of pro- and anti-inflammatory cytokines in epicardial adipose tissue and subcutaneous white adipose tissue in patients undergoing coronary artery bypass graft surgery. Immunohistochemical stain showed a reduced NF-кB activity and decreased macrophages infiltration and M1/M2 macrophages ratio in patients treated with nitrate esters. In line with this, the expression of pro-inflammatory cytokines was decreased, and that of anti-inflammatory cytokines was increased [19]. Whether these observations can be reproduced in larger studies remains to be tested. In sum, there is enough evidence to support the concept that organic nitrates have potentially deleterious effects on the vasculature, and particularly the endothelium. While the question of whether nitrate therapy modifies patients´ prognosis remains unanswered, it is highly unlikely that this type of therapy might have anti-atherosclerotic effects. 

## 5. Direct NO Donors

### Nicorandil

Nicorandil (Figure 3) is a nicotinamide ester with a dual pharmacological action, including the opening of ATP-sensitive potassium channels, which are prevalent in resistance arterioles, and an NO-mediated effect [20]. The IONA (Impact of Nicorandil in Angina) trial showed that nicorandil administration might lead to reduced incidence of coronary events in patients with stable angina during a mean follow-up period of 1.6 years [21]. Whether this effect is mediated by antiatherosclerotic properties of this molecule, or by its fibrinolytic effects, remains unclear. In the study by Izumiya et al., intravascular ultrasound-virtual histology was used to assess, in 65 patients with stable angina pectoris, whether therapy with nicorandil is associated with changes in plaque structure [22]. In this non-randomized, cross-sectional study, patients under nicorandil therapy showed more stable coronary plaques with features of fibrous tissue and a smaller necrotic core independently of statin use. In the same study, an 8-weeks nicorandil therapy reduced the area of atherosclerotic lesion and plaque necrosis in ApoE-deficient mice fed an atherogenic diet. Further, nicorandil significantly reduced the expression levels of endoplasmic reticulum stress markers, C/EBP homologous protein and glucose regulated protein/BiP in atherosclerotic lesions, and it significantly attenuated tunicamycin-induced CHOP upregulation in cultured THP-1 macrophages [22]. In sum, while the hemodynamic effects of this drug are potent (and associated with important side effects that limit its tolerability), the evidence on the antiatherosclerotic effect of this molecule is preliminary but potentially promising.

## 6. Molsidomine

Molsidomine and its active metabolite SIN-1 (Figure 4) are vasodilators and anti-platelet agents whose (NO-dependent) effects are devoid of tolerance and cross-tolerance phenomena [23,24]. The effect of molsidomine on carotid intimal hyperplasia has been tested in the ACCORD (Angioplastie Coronaire Corvasal Diltiazem) trial. In this study, 700 patients undergoing elective PTCA were randomized to receive either i.v. SIN-1 followed by oral molsidomine or calcium channel blockers. After 6 months of therapy, the molsidomine group showed a reduction in the rate of restenosis (an effect that was, however, not associated with a reduction in clinical events) [25]. Further, molsidomine therapy was associated with an improvement in flow-mediated dilation in patients with coronary artery disease [26], suggesting an improvement in the endothelial function whose mechanism remains however unclear. In another randomized study, including 50 patients undergoing coronary revascularization, 12 months of molsidomine administration failed to show a difference in endothelial biomarkers (sICAM, high sensitive-C reactive protein, myeloperoxidase oxidized-LDL) compared to placebo [27]. One marker, myeloperoxidase, which is linked to both inflammation and atherosclerosis [28,29], was reduced significantly in the molsidomine group. The effects of molsidomine and SIN-1 on the pathophysiology of atherosclerosis are much less well investigated. In a rabbit model, molsidomine (1 mg/kg/day) was not associated with plaque size changes, but it did modify plaque features, including increased thickness of the subendothelial macrophage-free layer and reduced superoxide formation, suggesting plaque stabilization [30]. In vitro and in patients with coronary artery disease [31], SIN-1 reduced the endothelial expression of adhesion molecules that favor adhesion and diapedesis of inflammatory cells into the plaque. Further, molsidomine administration was associated with a reduction in plasma levels of ICAM-1 [32]. In a recent paper by Roth et al. [33], the NO donor molsidomine administration in a murine model of atherosclerosis was associated with features of stability, including a thicker fibrous cap, increased occurrence of plaque macrocalcifications, reduced perivascular fibrosis. Even though survival was not affected, there was less evidence of neurological dysfunction, and the occurrence of myocardial infarctions was reduced significantly. Collectively, these findings confirm the beneficial effects of molsidomine on cardiac function, neurology, and atherosclerotic plaque stability; whether they will find application remains to be investigated.

## 7. Inorganic Nitrate and Nitrite

A number of strategies can be implemented to reduce the risk of cardiovascular events. Of these, one of the most effective ones is a diet rich in vegetables and poor in animal fats [34]. Although the benefit of this diet is clear from the epidemiological perspective, the nutrients and mechanisms responsible for this remain unclear. One plausible hypothesis for its cardioprotective effects is that these are mediated by their inorganic nitrate and nitrite content [35,36,37]. Upon assumption, inorganic nitrate/nitrites (NOx) are reduced to nitrites by the oral flora, which is reduced to NO by the gastric acids. NO can, in turn, cross the gastric mucosa into the portal circulation, be again oxidized to the more stable nitrite in the liver, and then again reduced to NO in the peripheral circulation, a reaction that is favored in hypoxemic environments. This latter characteristic allows preferential NO release in ischemic areas, making the vasodilatory effects of this NO source more selective. 

A number of protective cardiovascular effects have been shown for inorganic nitrite and nitrates [38,39,40]. In animal models, these include (among others) a reduction in circulating biomarkers of oxidative stress, improved myocardial remodeling, improved vascular NO and cGMP levels, increased eNOS function as well as inhibition of the NADPH oxidase isoform 4, leading to improved macro- and microvascular responsiveness. From the metabolic perspective, inorganic nitrate therapy is associated with a reduction in triglyceride plasma levels, blood glucose levels, and glycated hemoglobin values. Inorganic nitrite/nitrate therapy has also shown anti-inflammatory properties, as manifested by reduced tissue levels of tumor necrosis factor-alpha and adhesion molecules and reduced leukocyte recruitment [27]. In a murine model of hind-limb ischemia, a 14-days dietary nitrate supplementation was associated with increased plasma S-nitrosothiols and nitrite, preserved perfusion to ischemic limbs, increased mobilization of angiopoietic cells, and reduced myocyte apoptosis. These protective effects were lost when nitrate-reducing bacteria in the oral cavity were removed by regular use of an antiseptic mouthwash [28]. Further, inorganic nitrite/nitrate also showed a plaque-stabilizing effect, as manifested by reduced leucocytes adhesion, reduced recruitment to the vascular wall, and reduced myeloperoxidase activity [29]. These changes were associated with reduced macrophage content and increased smooth muscle cell content of atherosclerotic plaques, i.e., in a more stable plaque phenotype. Of note, the above observations were not reproduced in other animal models [41]. Sindler et al. reported an improvement in age-associated endothelial dysfunction in animals fed a nitrite-rich diet, with evidence of reduced arterial stiffness and reduced levels of oxidative stress [42].

Finally, inorganic nitrates have been associated with reduced infarct size following experimental myocardial ischaemia, a phenomenon similar to the so-called ischemic preconditioning. This effect appears to be present for doses of inorganic nitrite up to 4mg/kg, but not higher, and it is lost when the nitrite is administered at the end, but not before, ischemia [43]. Importantly, this effect was also reproduced in patients with ST-elevation myocardial infarction, who showed an improvement in the myocardial salvage index and a reduced incidence of events during a 1-year follow-up in the group that received inorganic nitrite infusion intracoronarily during the percutaneous treatment [44]. Inorganic nitrate/nitrite has also been shown to possess platelet inhibitory effects: in vitro, inorganic nitrite causes cGMP-mediated inhibition of human platelet aggregation, while inorganic nitrate causes an increase in circulating nitrite levels and nitrate inhibits platelet function in eNOS(−/−) mice in a in vivo model of thromboembolic radiolabeled platelet aggregation [45]. In other words, nitrite and nitrate appear to compete and balance endogenous NO, and to be particularly efficient in settings of ischemia. 

In humans, plasma levels of inorganic nitrate have been reported to correlate with endothelial vasomotor function [46] and improved blood pressure. In a randomized cross-over trial, vascular conductance and endothelial function improved after nitrate administration as compared to placebo. The changes in systolic blood pressure and endothelial function induced by nitrate were, however, inversely proportional to the respective baseline values observed in the placebo condition, which suggests that nitrate supplementation might be an adjuvant, but not a therapy, for hypertension [47]. While the acute effects of nitrite have been well investigated, the impact upon chronic assumption is less clear. This endpoint is currently the subject of investigation of the NITRATE-TOD trial [48]. Daily intake of inorganic nitrates improves flow-mediated dilation in some, but not all studies [38]. Finally, in a recent meta-analysis, inorganic nitrate intake led to significantly reduced resting blood pressure (systolic blood pressure: −4.80 mmHg, *p* < 0.0001; diastolic blood pressure: −1.74 mmHg, *p* = 0.001), to an improved endothelial function (flow-mediated dilatation: 0.59%, *p* < 0.0001), to a reduction in arterial stiffness (pulse wave velocity: −0.23 m/s, *p* < 0.0001; augmentation index: −2.1%, *p* = 0.05), and to a reduced platelet aggregation by 18.9% (*p* < 0.0001) [43]. In humans, the effects of inorganic nitrites and nitrates on atherogenesis remain to be investigated, although dietary nitrate has been reported to improve exercise performance in patients with peripheral artery disease [49]. 

## 8. Drugs that Act on the NO Pathway

### 8.1. eNOS Activators

As discussed above, there is now consistent evidence supporting an important pathophysiological role of NOS dysregulation/uncoupling in atherosclerotic diseases. Pharmacological targeting of this enzyme, in particular the development of strategies aimed at “recoupling” its function, has, therefore, potential implications for atherogenesis. So-called “NOS enhancers”, i.e., molecules that upregulate the expression of NOS at the mRNA and protein level, have been developed [50,51,52]. In the study by Wohlfahrt et al., male apoE-KO mice were randomized to receive either standard rodent chow or chow supplemented with AVE9488 for 12 weeks. This eNOS activator reduced cuff-induced neointima formation and atherosclerotic plaque formation in apoE-KO mice (an effect that was absent in apoE/eNOS-double knockout mice) [52]. The effect of the same molecule was recently tested on cardiac ischemia/reperfusion injury in an in vivo murine model. Treatment with the eNOS enhancer AVE9488 (30 mg/kg/day) for one week was associated with a significant reduction in the ischemic area as compared to placebo (infarct/area at risk 65.4 +/− 4.1 vs. 36.9 +/− 4.0%, *p* = 0.0002). This effect, which was accompanied by a reduction in markers of oxidative stress, was blunted in eNOS knockout mice (infarct/area at risk 64.1 +/− 6.2%) [53]. In analogy, treatment with the same compound led to a more favorable left ventricular remodeling in a similar model of ischemia [50]. In another study, the eNOS enhancer AVE3085 was tested in a model of porcine coronaries exposed to homocysteine. Exposure to this risk factor worsened endothelial function and NO release, downregulated eNOS mRNA expression and protein expressions of eNOS and p-eNOS(Ser1177), while it upregulated iNOS expression, possibly as a marker of inflammation. In contrast, AVE3085 restored NO release and endothelium-dependent relaxation while decreasing iNOS expression. These effects were mediated by activation of protein kinase Akt and PI3 kinase [54]. Regarding the potential impact of these molecules on atherogenesis, no data are currently available. 

Although this seems to be an important strategy, the fact that post-translational modifications (i.e., S-glutathionylation) and/or oxidation of the co-factor BH_4_ uncouple the enzyme makes an approach based on the overexpression alone of this enzyme not sufficient. In the study by Wohlfahrt et al., indeed, there was evidence of AVE9488-induced eNOS uncoupling [52]. Further, to date, these concepts have not been tested in humans. With regard to the exogenous supplementation of BH_4_ alone, or to strategies aimed at supporting the regeneration of this cofactor, direct BH_4_ treatment or indirect via folic acid supplementation has been associated with improved endothelial function in a number of patient groups with cardiovascular disease, e.g., chronic smokers, diabetes, and patients with nitrate tolerance [18,55,56]. Results from a meta-analysis show that folic acid supplementation reduces the progression of intima-media thickness, particularly in subjects with chronic kidney disease or high cardiovascular risk factors burden [57], even though negative data also exist [32]. In a recent randomized, prospective study, Nygard et al. investigated 102 patients with stable angina pectoris who were randomized to receive folic acid/vitamin B_12_ or placebo for 1 year after undergoing percutaneous coronary intervention. The authors investigated the effect of the therapy on coronary atherosclerosis using an ultrasound-based technology (virtual-hystology) which allows, based on the change in frequency of the reflected waves, to distinguish features of unstable (necrotic core, thin fibrous cap) from features of stable (fibrotic plaque) coronary lesions. The presence of thin-cap fibroatheromas was associated in these patients with elevated values of MCP-1, a chemoattractant that reflects inflammation and plaque activation. In these patients, folic acid supplementation was not associated with either presence of thin-cap fibroatheromas or the levels of MCP-1 [58].

### 8.2. Soluble Guanylate Cyclase Activators

A number of more modern pharmacotherapies were designed to modify the NO-cGMP axis; their impact on atherogenesis is, however, incompletely investigated. Soluble guanylate cyclase (sGC, the target of NO) activators increase NO downstream signaling and directly address one of the potential sources of impaired activity of the NO-axis since oxidation of the sGC is also a mechanism of inactivation of this enzyme. sGC stimulators and activators, whether dependent or independent of NO, have been developed for the treatment of pulmonary hypertension, chronic thromboembolic pulmonary hypertension, and heart failure [59,60]. Among these molecules, cinaciguat (sGC activator) improves vascular oxidative stress and coronary endothelial function in a rodent model of myocardial infarction [61]. 

### 8.3. Phosphodiesterase Inhibition

Finally, inhibitors of phosphodiesterases (PDE) increase cGMP levels and are used in multiple indications. They have been shown to improve endothelial responsiveness [61,62]. Another study from the same laboratory using Apo^−/−^ mice showed a 40% reduction in aortic plaque deposition with 3 months Sildenafil (Viagra^®^, 40 mg/kg/day, orally) treatment [63]. However, more translational research is needed [62]. 

### 8.4. Arginase Inhibition

Arginase is a manganese metalloenzyme that catalyzes the hydrolysis of L-arginine to urea and L-ornithine and can be found in multiple cell types, including vascular cells and circulating red blood cells. The depletion of L-Arg leads to NOS uncoupling, with O_2_ rather than L-Arginine as the terminal electron acceptor, resulting in superoxide formation. In a recent in vitro study, L-arginine values of 100 µM less led to a progressive increase in peroxynitrite and superoxide anion, an effect that was blocked by a NOS inhibitor, compatible with NOS uncoupling [63]. Arginase is activated by a variety of inflammatory mediators, resulting in reduced bioavailability of L-arginine NO. Recently, Arginase gained interest as a potential target to improve NO production [64,65]. In a study on non-senescent human endothelial cells, overexpression of Arg-I led to eNOS-uncoupling, expression of senescence markers and inflammatory markers as well as monocyte adhesion to endothelial cells via increased oxidative stress [66]. In line with this, oxidative stress in older animals was associated with elevates Arginase 1 and eNOS uncoupling [67].

In contrast, downregulation of arginase activity results in reduced eNOS uncoupling, reduced ROS production due to reduced substrate (*L*-arginine) deficiency, and, accordingly, in enhanced NO bioavailability and improved endothelial function [68].

In atherogenic mice (ApoE^−/−^) fed with a high-cholesterol diet, an 8-week treatment with an arginase inhibitor *S-(2-boronoethyl)-L-cysteine* reduced aortic plaque burden [69], highlighting the potentially important role of this agent in atherosclerosis. Although the bulk of current knowledge on arginase inhibition is based on animal studies, results of initial clinical studies support the feasibility of this approach also in humans. One hundred and twenty minutes of intra-arterial infusion of the arginase inhibitor *N-hydroxy-nor-L-arginine* (nor-NOHA; 0.1 mg/min) markedly improved endothelial function in patients with coronary artery disease and diabetes mellitus and familial hypercholesterolaemia [70,71]. However, given the central role of arginase in the urea cycle, future research should be focused on isoform- and tissue-selective agents, on the identification of the optimal dosing and on strategies aimed at selectively targeting the vascular effects of these therapies.

## 9. Conclusions

The vascular endothelium and its product NO are important mediators of vascular health. Reduced bioavailability of NO is associated with increased platelet activity, reduced vasomotion, inflammation, and with the initiation, progression, and complications of atherosclerosis. Several NO donors have shown an effect on multiple pathways that mediate inflammation and atherogenesis (Table 1). Nonetheless, exogenous NO substitution still has not found a clear application, except for the use of organic nitrates or other NO donors in the prevention/treatment of angina and the acute therapy of congestive heart failure. Despite positive evidence from animal studies, the potential of the administration of direct or indirect NO donors, the enhancement of the production of NO, as well as of strategies aimed at sensitizing the cGMP pathway, remain unexploited. Translational studies in this direction are now needed. 

## Figures and Tables

**Figure 1 ijms-21-02703-f001:**
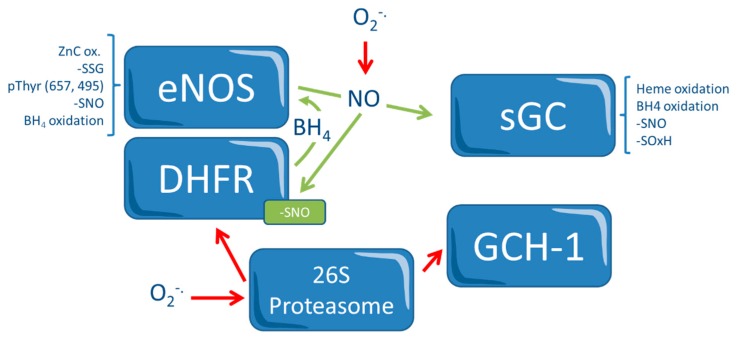
Redox-control in the NO Pathway. Redox-dependent pathways that regulate nitric oxide (NO) and vascular homeostasis. The endothelial nitric oxide synthase (eNOS) and the soluble guanylyl cyclase (sGC) present a number of redox switches, which act directly or indirectly (for instance, in the case of the 26S proteasome, which is activated via 3-nitrotyrosine (3NT) modification causing inactivation of the eNOS cofactor tetrahydrobiopterin (BH_4_) synthase GTP-cyclohydrolase (GCH-1), and of BH4 recycling enzyme dihydrofolate reductase (DHFR). Oxidation of BH_4_ causes a positive feedback mechanism leading to further oxidative stress. Red arrows indicate inhibition. In contrast, NO prevents proteasomal DHFR degradation via tyrosine nitration of the 26S proteasome. O2^−^: superoxide anion; -SNO: s-nitroso-thiols; -SOxH: oxidized thiol; -SSG: glutathione disulfide; ZnC: Zinc; pThyr: phosphorylated thyrosine.

**Figure 2 ijms-21-02703-f002:**
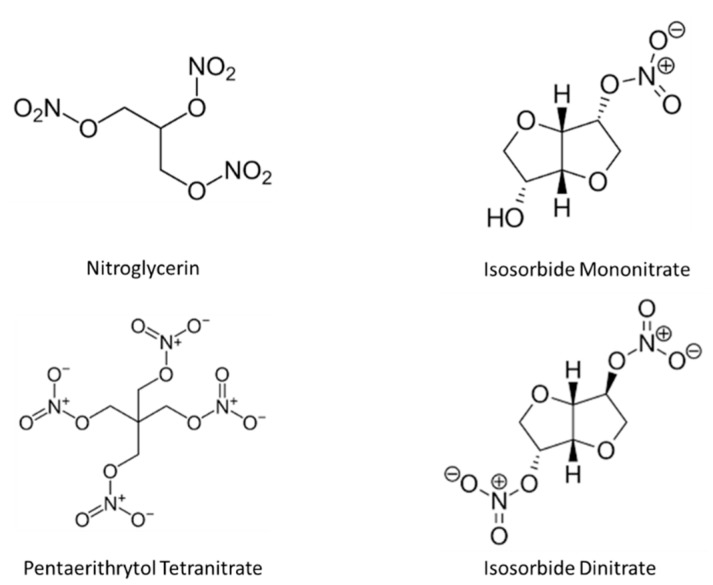
Organic nitrates.

**Figure 3 ijms-21-02703-f003:**
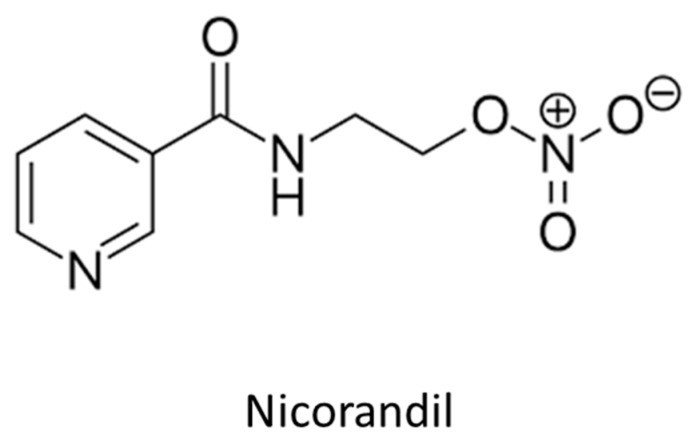
Nicorandil.

**Figure 4 ijms-21-02703-f004:**
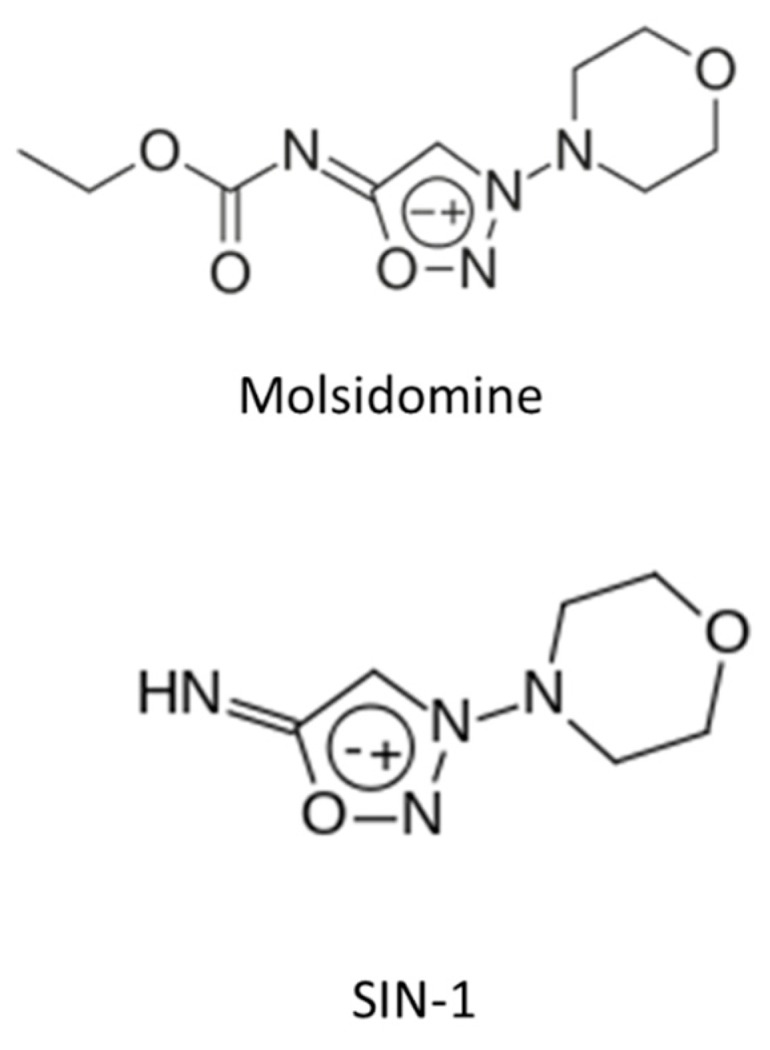
Molsidomine and SIN-1.

**Table 1 ijms-21-02703-t001:** NO donors and their effects on cytokines.

Drug	Effect	Reference
**Nitrate esters**	Inhibition of NF-kappaB-regulated macrophage polarization shift	[19]
**SIN-1**	Inhibition of IL-1 beta-mediated induction of intercellular adhesion molecule-1 (ICAM-1) and VCAM-1 expression	[31]
**Nitroxyl anion and nitroglycerin**	Reduction of intercellular adhesion molecule-1 (ICAM-1) and the cytokines monocyte chemoattractant protein 1 (MCP-1) and interleukin 6 (IL-6)	[72]
**Nitroglycerin**	Reduced plasma levels of C-reactive protein and sE-selectin	[73]
**Nitroglycerin**	No change in asymmetric dimethyl-arginine (ADMA), human soluble P-selectin, interleukin-6, tumor necrosis factor-alpha, intercellular adhesion molecule-1, and oxidized low-density lipoproteins in healthy volunteers	[74]
**Nicorandil**	Increased Mst1 and p-Mst1 levels in cardiomyocytes subjected to MI injury	[75]
**Nicorandil**	Inhibition of Toll-like receptor 4 (TLR4)-mediated myeloid differentiation primary response protein 88 (MyD88)-dependent nuclear factor-kappa B (NF-κB) signaling	[76]
**Nicorandil**	Inhibition of cyclic strain-induced IL-8 expression	[77]
**Nicorandil**	Reduced expression of 53 cytokines after percutaneous intervention in patients with coronary artery disease	[78]
**Nicorandil**	Inhibition of TNF-alpha in vitro	[79]
**Nicorandil**	Inhibition of the activation of NF-kappaB, expression of adhesion molecules, and cytokine production	[80]
**Nicorandil**	Inhibition of TNFalpha release from lymphocytes	[80]
**Inorganic nitrate**	Reduced systemic leukocyte rolling and adherence, circulating neutrophil numbers, neutrophil CD11b expression and myeloperoxidase activity	[29]

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
