# Peer review of "Exogenous NO Therapy for the Treatment and Prevention of Atherosclerosis"

_ijms, 2020, doi:10.3390/ijms21082703_

Round 1
Reviewer 1 Report
The authors have detailed a concise list of agents that act on NO pathway and thus propose to be employed for the treatment of atherosclerosis. I appreciate the authors for covering this important topic. However, I believe the article should provide more detail regarding the role of each agent on the cytokines involved in vascular homeostasis in order to fully appreciate their potential effects in atherosclerosis. Below are some of my comments to improve the review further.
Minor comments:
- Words missing in line 31.
- Expand all the acronyms from figure 1 in the figure legend.
Major comments:
- I would suggest representing NO pathway using a more elaborate figure
- When discussing each agent, mention which part of the NO pathway they interact with, also add them in the figure (maybe with an arrow) showing where they exactly act upon.
- I would suggest adding a table to describe the role of each specified NO agents on the cytokines involved in vascular homeostasis and thereby atherosclerosis.
Author Response
Reviewer 1
The authors have detailed a concise list of agents that act on NO pathway and thus propose to be employed for the treatment of atherosclerosis. I appreciate the authors for covering this important topic. However, I believe the article should provide more detail regarding the role of each agent on the cytokines involved in vascular homeostasis in order to fully appreciate their potential effects in atherosclerosis.
This is a very good comment. I have tried to describe these pathways more in detail, when available (see below).
Below are some of my comments to improve the review further.
Minor comments:
- Words missing in line 31.
Thank you, this sentence has been changed.
- Expand all the acronyms from figure 1 in the figure legend.
This has been done, thank you.
Major comments
1. I would suggest representing NO pathway using a more elaborate figure
Thank you very much for this comment. I have expanded the central figure to include more details on the pathways of the NO axis (Ca2+ uptake, mechansims of NO production etc) and all drugs discussed in the review. I am actually not sure whether the central figure is visible during the review process. According to authors´guidelines, it should not be included in the text.
2. When discussing each agent, mention which part of the NO pathway they interact with, also add them in the figure (maybe with an arrow) showing where they exactly act upon.
You are right. This has been more clearly emphasized in the central figure. I have pasted the central illustration at the end oft he manuscript to make sure the reviewer can evaluate it.
3. I would suggest adding a table to describe the role of each specified NO agents on the cytokines involved in vascular homeostasis and thereby atherosclerosis.
Thank you very much for this comment. I have introduced a Table listing the evidence regarding the impact of different NO-based agents on cytokines. I hope you will find this satisfactory.
Reviewer 2 Report
In his review article: „Exogenous NO therapy for the treatment and prevention of atherosclerosis“, Tommaso Gori gives a precise overview of compounds incorporated in nitric oxide pathways or producing NO themselves with focus on side-effects on the formation or prevention of atherosclerosis. The article is well-written and unveils a good structure as well as broad literature research. It is of relevance for readers in and outside the field.
I only have some minor remarks:
- How informative are the chemical structures? I understand, the article would miss figures otherwise. But, I personally do not see a benefit.
- Please avoid persuasions like “most interesting one…” (line 74) and stay objective.
- Spelling mistakes in lines 133 (was), 245 (meta-aanlysis), 288 (study study).
Author Response
Reviewer 2
In his review article: „Exogenous NO therapy for the treatment and prevention of atherosclerosis“, Tommaso Gori gives a precise overview of compounds incorporated in nitric oxide pathways or producing NO themselves with focus on side-effects on the formation or prevention of atherosclerosis. The article is well-written and unveils a good structure as well as broad literature research. It is of relevance for readers in and outside the field.
I only have some minor remarks:
- How informative are the chemical structures? I understand, the article would miss figures otherwise. But, I personally do not see a benefit.
Thank you for this comment. The molecular structure of the compounds was actually not included in the original version of the paper, it was added upon request by the editors. I leave the decision to the editors and am available for any change.
2. Please avoid persuasions like “most interesting one…” (line 74) and stay objective.
You are absolutely right, this has been changed throughout to the advantage of the paper. I found the term „interesting“ in at least 6 different places, which I agree was absolutely too many. I fully concur with the point of the reviewer.
3. Spelling mistakes in lines 133 (was), 245 (meta-aanlysis), 288 (study study).
Thank you very much, I apologize for the mistakes.
Round 2
Reviewer 1 Report
I appreciate the authors for making the relevant changes.